# Can Pest Management and Cultivar Affect *Phytoptus avellanae* Infestations on Hazelnut?

**DOI:** 10.3390/insects15100740

**Published:** 2024-09-26

**Authors:** Mario Contarini, Roberto Masturzi, Eleonora Iezzi, Miloš Petrović, Cristian Silvestri, Silvia Turco, Stefano Speranza, Luca Rossini

**Affiliations:** 1Dipartimento di Scienze Agrarie e Forestali, Università degli Studi della Tuscia, Via San Camillo de Lellis snc, 01100 Viterbo, Italy; roberto.masturzi@unitus.it (R.M.); eleonora.iezzi@unitus.it (E.I.); silvestri.c@unitus.it (C.S.); silvia.turco@unitus.it (S.T.); 2Department for Environmental and Plant Protection, Faculty of Agriculture, University of Novi Sad, Trg Dositeja Obradoviča 8, 21000 Novi Sad, Serbia; milos.petrovic@polj.edu.rs; 3Centro de Estudios Parasitológicos y de Vectores (CEPAVE, CONICET-UNLP), La Plata B1900, Argentina; 4Service d’Automatique et d’Analyse des Systèmes, Université Libre de Bruxelles, Av. F.D. Roosvelt 50, CP 165/55, 1050 Brussels, Belgium; luca.rossini@ulb.be

**Keywords:** Eriophyoidea, *Corylus avellana*, cultivar susceptibility, big bud mite, phytosanitary management

## Abstract

**Simple Summary:**

Hazelnut cultivation is quickly expanding worldwide due to increasing hazelnut demand. Despite its resilience and tolerance to many abiotic and biotic stresses, *Corylus avellana* expansion is continuously exposing the crop to new adversities, while climate change is causing the resurgence of many native pests and diseases. The mite pest *Phytoptus avellanae* is responsible for substantial yield reductions in many productive areas. Cultivar and pest management strategies can affect the incidence and the population dynamics of this pest, but to date understanding remains limited. This lack of knowledge inspired the present study.

**Abstract:**

The big bud mite *Phytoptus avellanae* is a resurgent pest of hazelnut, *Corylus avellana*, causing substantial yields reductions in many productive areas. Mites colonise and develop within healthy buds which become swollen, with subsequent alteration to the plant’s development. To date, there has been limited knowledge on how the cultivar and pest management strategies affect infestations. This study explored these aspects through two ad hoc experiments carried out in central Italy. In the first experiment, the susceptibility of 11 cultivars with different geographic origins was tested in a germplasm hazelnut collection. The second experiment assessed the infestation level in orchards with integrated pest management (IPM) and organic pest management strategies and in a renaturalised environment (a former agricultural area now converted in a natural park). The results showed that the most and the least susceptible cultivars were Tonda Gentile and Nocchione, respectively. No significant differences were found between IPM and organic management, but they were both different to the renaturalised environment. The outcomes of this research can serve as a valuable reference and can be applied to all current or potential hazelnut cultivation areas characterised by the same environmental conditions.

## 1. Introduction

Hazelnut, *Corylus avellana* L., is an important crop valued for its organoleptic properties, nutritional benefits, and the nutrient bioavailability of its fruits, the nuts [1]. The growing interest in this crop has led to an expansion from Turkey, Italy, and Spain, the main producers [2], to other Mediterranean countries, the US, and to the southern hemisphere, including Chile and Australia [3]. Hazelnut is well known as a plant tolerant to numerous abiotic (e.g., drought, high temperatures) and biotic stressors (e.g., pests and diseases); nonetheless, in recent years many challenges have emerged that threaten to reduce yields [4,5,6,7,8,9,10]. Moreover, climate change, influencing hazelnut growth and yield, may also affect the future of this crop, although the severity of these environmental alterations is still under evaluation. For instance, it may lead to an expansion towards other suitable areas, or to a substantial reduction in the yields and the organoleptic properties of nuts in the warmest areas [11]. The scientific community, especially breeders, are prioritising the selection of tolerant varieties which can offer constant and high-quality yields, low susceptibility to pests and diseases, and tolerance to extreme weather events [12,13]. Besides the practical and economic advantages for farmers’ business, this research contributes to a substantial reduction in management inputs (e.g., energy, water, agrochemicals) in cultivated fields, leading to more sustainable production. Conversely, the phytosanitary management of hazelnut orchards plays a crucial role in shaping the environmental impact of this crop. Integrated pest management (IPM) and organic farming are two common management practises. While in organic fields synthetic pesticides are not allowed, in IPM they are limited by the restrictive regulations released by regional and national phytosanitary authorities [14,15,16]. Consequently, the protection of hazelnut orchards is becoming increasingly complex, above all in case of severe outbreaks.

Mites belonging to the Eriophyoidea superfamily, *Phytoptus avellanae* Nalepa (Acari: Phytoptidae) *lato sensu* [17,18] and *Cecidophyopsis vermiformis* (Nalepa) (Acari: Eriophyidae), are perfect examples of the biotic stressors that are an obstacle in sustainable farming. These eriophyoids infest vegetative and reproductive buds, causing an alteration in the colonised plant tissues that leads to the formation of pseudo-galls, also named “big buds”. A direct consequence of the mites’ activity is the reduction in the number of buds, which leads to delayed plant development and reduced nut production [17]. Both species can be found within big buds, but the role and the presence of *C. vermiformis* is still partially unclear and deserves further studies. In contrast, the biology of *P. avellanae* is better described, and it is considered the main species responsible for production losses [19]. Symptoms of mite infestations are easily distinguished, especially during the winter months, as the infested buds become swollen and reddish [20]. During leaf outbreak, they do not open and can be easily distinguished from healthy buds, as they do not develop into new shoots. These properties of the infested buds make winter a favourable period of the year to assess the infestation level in hazelnut orchards [21].

Over the years, big bud mite infestations have been controlled using agrochemicals such as endosulfan [22], but their impact on humans and the environment has led to their exclusion from pest control programs. To date, the application of sulphur-based products remains a standard strategy for controlling mite outbreaks, either in IPM or organic hazelnut orchards [23]. Recently, research on mite infestation leveraged advanced genomic techniques to gain a deeper understanding of the microbial community potentially involved in the processes leading to gall formation [15]. The increasing “omics” research and the genetic studies focused on the selection of more tolerant cultivars, along with the unsolved problems associated with pest management [12,15,16], were the main source of inspiration for this study. In order to better understand the susceptibility of different hazelnut cultivars grown under the same environmental conditions, together with the possible involvement of orchard management in outbreaks, two independent monitoring experiments were conducted over two consecutive growing seasons (2023–2024).

The first experiment was carried out in a collection orchard located in Caprarola, in the “Viterbo hazelnut district” (central Italy), to assess the infestation level of different hazelnut cultivars growing under the same climatic and management conditions. The second experiment was carried out in hazelnut orchards located in the same productive area, managed with three different practises: IPM, organic, and renaturalised. Notably, the latter is a peculiar environment where the cultivations were abandoned thirty years ago, and that over time gave birth to a natural ecosystem that is today enclosed in a natural park. Together, these two experiments shed new light on the genetic and agronomic implications of big bud mite infestation and provide useful information for sustainable hazelnut orchard management.

## 2. Materials and Methods

### 2.1. Experimental Orchards

#### 2.1.1. Hazelnut Varieties’ Susceptibility: Germplasm Collection Site

The evaluation of the susceptibility of different hazelnut cultivars was carried out at “Le Cese” (42°20′53″ N 12°11′38″ E) orchard, located in the municipality of Caprarola (Viterbo, Latium, central Italy). “Le Cese” is a hazelnut germplasm collection established in 2000 by the Regional Agency for Agricultural Development and Innovation in Latium (ARSIAL) with the aim of evaluating and comparing the vegetative and productive performances of 49 hazelnut varieties of different origins and commercial relevance worldwide. Each variety is represented by three adjacent plants. The orchard has a size of 0.4 hectares and consists of adult plants, trained as multi-stemmed open vase, organised in a regular layout of 5 × 4 m. The orchard is subject to traditional agronomic management practices consisting of standard fertilisation, irrigated through a sub-irrigation system, light winter pruning, and the absence of phytosanitary treatments. A naturally occurring green cover crop is used to manage the soil. The average plant height depends on the cultivar, but generally it ranges between 3 and 5 m.

#### 2.1.2. Different Pest Management Sites

To evaluate the incidence of big bud mite infestations in IPM, organic, and renaturalised environments, we selected different fields located in the Viterbo hazelnut district (Latium, central Italy) (Table 1). IPM orchards are managed according to phytosanitary guidelines issued by the Latium Region [24], which regulate the number of treatments and the active ingredients that can be applied. Organic hazelnut orchards were selected among the farms that have been certified by official independent bodies. The renaturalised sites were located in the Regional Natural Park “Valle del Treja”, in hazelnut orchards where cultivation was interrupted during the 1990s and where, to date, hazelnut trees live in association with trees (i.e., *Quercus* spp., *Fraxinus ornus*, *Ostrya carpinifolia*, *Ulmus minor*, *Acer* spp.), shrubs (i.e., *Rubus* spp., *Arbutus unedo*, *Rhamnus* spp., *Ruscus aculeatus*, etc.), and herbaceous species (i.e., *Ranunculus* spp., *Pteridium aquilinum*, *Cyclamen* spp., *Rubia peregrina*, etc..) which altogether have created a natural ecosystem and constitute a highly diversified forest stand with plant species typical of the Mediterranean context.

The survey of the growing season 2023 was conducted on 3 sites for each management system. However, three more sites were added for the IPM and organic environments in 2024, while the number of renaturalised sites remained unvaried, due to their uniqueness. In 2024, one organic orchard (Sutri) was converted to IPM and was replaced by an additional one (Vetralla 2).

Hazelnut orchards were selected based on certain common characteristics: hazelnut plants belonged to the cultivar Tonda Gentile Romana, trees were more than 10 years old and in their full production phase, and permanent natural ground cover was adopted, with mowing operations that become more frequent near the harvest to make this task easier. The identification of the cultivar in the renaturalised sites, instead, was carried out firstly by collecting historical information and then on site by morphological identification [25].

The survey involved irrigated and non-irrigated orchards, as previous studies [21] have shown that there is no effect of water supply during the summer on big bud mite infestation levels.

### 2.2. Monitoring Activity

Monitoring activities were conducted in January 2023 and 2024, when plant dormancy makes the discrimination between infested and non infested buds easier. In the germplasm collection orchard, we selected the 11 varieties (Table 2) that were represented by all three hazelnut plants in healthy condition, discarding the 38 varieties that did not fulfil these requirements.

In the second experiment, instead, we randomly selected 10 plants per each orchard listed in Table 1 that represented the stand well.

The monitoring protocol was the same for the two experiments and it was also the same as we applied in previous studies [21]. Specifically, on each plant, we randomly selected five branches; then, on each branch, we counted 50 buds from the distal part and we recorded the number of healthy and infested buds.

### 2.3. Data Analysis

The data collected were imported into the R environment [26] and analysed following the scripts publicly available at https://github.com/lucaros1190/Phytoptus2024LeCese (accessed on 1 September 2024).

#### 2.3.1. Different Susceptibility of the Cultivars

Data were analysed through a Generalised Linear Model with random effect (GLMER) with a negative binomial distribution and the Bonferroni post hoc test (=0.05). The analysis was carried out through the glmer.nb() function within the R package lme4 [27], followed by a pairwise comparison carried out through the emmeans() function within the R package emmeans [28], the pairs() function within the R package multcompView [29], and the cld() function within the R package multcomp [30]. In this analysis, we considered the number of big buds per branch as a dependent variable, the growing season and the cultivars as independent variables, and the plant and the branches as random factors.

#### 2.3.2. Different Susceptibilities among the Orchard Management Systems

This part of the analysis was carried out in two steps. The first step focused on the overall dataset (season 2023 and 2024 altogether) to analyse the differences in the infestation level between the two growing seasons and among the different types of orchard management (IPM, organic, and renaturalised). Data were analysed through a GLMER with a negative binomial distribution and the Bonferroni post hoc test (=0.05), using the same R packages and functions listed in Section 2.3.1. As previously performed, the number of big buds per branch was considered as a dependent variable, the growing season and orchard management as independent variables, and the orchard, the plant, and the branches as random factors.

The second step of the analysis was considered as a double check and, analogously to Section 2.3.1, the procedure was the same but did not consider the years as independent variables.

## 3. Results

### 3.1. Different Susceptibility of the Cultivars

The infestation level, expressed as the number of big buds per branch, assessed at the “Le Cese” varietal collection did not detect differences between the two years of observations (GLMER, Z = 0.37, *p* = 0.70). The comparison of the infestation level among the 11 cultivars showed significant differences, identifying groups that are more and less susceptible to the activity of the mites (Figure 1, Table A1).

The lowest infestation level was seen for the cultivar Merveille de Bollwiller, while the highest one was seen for the Tombul cultivar. Merveille de Bollwiller, San Giovanni, and Nocchione were not statistically different from each other and from Tonda di Giffoni, Barcelona, Negret, Ennis, and Riccia di Talanico; conversely, they have lower levels of infestation than Tombul, Tonda Gentile Romana, and Tonda Gentile. Tonda di Giffoni, Barcelona, Negret, Ennis, and Riccia di Talanico were not statistically different from Tombul and Tonda Gentile Romana but were less infested than Tonda Gentile.

### 3.2. Different Susceptibility among the Orchard Managements

Different to that reported in Section 3.1, the infestation level, in terms of number of big buds per branch, was significantly higher in the 2023 growing season than 2024 (GLMER, Z = 3.03, *p* = 0.0025). In the 2023 growing season, a higher infestation level was observed in the IPM fields, followed by the organic and renaturalised environments (Figure 2A). The infestation level in the renaturalised orchards was significantly different from the IPM (GLMER, Z = 5.23, *p* < 10^−4^) and organic (GLMER, Z = 4.72, *p* < 10^−4^) fields, while no statistical differences were observed between the IPM and organic (GLMER, Z = −0.58, *p* = 1) environments. A similar scenario in terms of differences was observed for the 2024 growing season (Figure 2B), but in this case the highest infestations were observed in organic fields, followed by the IPM and renaturalised fields. Despite this inversion, the organic and IPM environments were not statistically different from each other (GLMER, Z = 1.00, *p* = 0.96), but both were statistically different from the renaturalised environment (GLMER, Z = 4.41, *p* = 10^−4^; and Z = 3.67, *p* = 7 × 10^−4^, respectively).

## 4. Discussion and Conclusions

This two-year study has highlighted two key aspects that can influence big bud mite infestations in hazelnut orchards: varietal susceptibility and a phytosanitary management system. To the best of our knowledge, it is the first time that such a study has been carried out in central Italy, where the hazelnut is a historical and expanding cultivation [31]. However, these results can serve as a valuable reference and can be applied to all current or potential hazelnut cultivation areas, at least those characterised by the same environmental conditions. *Phytoptus avellanae* infestations are well known in many areas where the hazelnut is cultivated, but, in recent years, outbreaks have been rising to alarming levels, probably due to the gradual increase in temperatures and the expansion of the crop towards areas not particularly suitable for growth [32]. This is the reason why deeper knowledge on how farm management practises and cultivars may affect infestations of big bud mites is crucial for more sustainable and low-impact production.

The results of the two experiments highlighted several interesting points that are worthy of discussion. Plant susceptibility to the action of *P. avellanae* is rooted in the genotypic variability of the cultivars, at least if they are grown under the same environmental conditions. Our results highlighted a low susceptibility of cultivars such as Merveille de Bollwiller, Nocchione, and San Giovanni, confirming what was already reported by Marinoni et al. [33] and Catarcione et al. [34]. The number of big buds in the three cultivars was significantly lower than the values observed of Tombul and Tonda Gentile, largely cultivated in some of the most important hazelnut districts of the world and widely appreciated for their organoleptic characteristics. The high susceptibility of Tombul and Tonda Gentile to the activity of big bud mites is not new, as was already observed by Stamenkovic et al. [18] and Tuncer et al. [35], and this study further confirmed this trait. There are many aspects that could explain this phenomenon, including the climatic conditions, the soil where the trials were carried out, and the specific susceptibility of the cultivar, confirmed in our study, to *P. avellanae*. Accordingly, the natural presence of pests in the areas where collection fields are established is an interesting aspect that should be taken into account in the analysis of between planted cultivars.

Tombul is the most widespread cultivar in Turkey, the world’s largest producer of hazelnuts. The average hazelnut yield in Turkey, however, is lower than in other countries, mainly because of the traditional propagation, cultivation practises, reduced mechanisation, and the impact of biotic adversities, including the big bud mite [36,37]. The improvement and development of breeding techniques, as reported by Oztolan-Erol et al. [37], to select genetic lines less susceptible to big bud mite infestation will be crucial to increasing Turkish hazelnut production. Tonda Gentile, instead, is one of the most appreciated hazelnut cultivars worldwide, especially because of the organoleptic quality of its nuts, widely requested by high-end confectionery product industries [38]. For this reason, it has been introduced to several countries outside Piedmont (north Italy), its natural distribution range, such as Japan and Chile, where hazelnut production is an emerging reality [39,40].

Adaptation to different environmental conditions, even if still suitable for hazelnuts, is a crucial point and this is the reason why collection fields such as “Le Cese” are being established in different areas [41]. The main utility of collection fields is to understand the behaviour of the different cultivars, in terms of phenological response, production, and susceptibility to biotic and abiotic stresses in general [42]. Regarding hazelnut cultivation, this interest has led to studies similar to that of Pacchiarelli et al. [43], where, among various findings obtained, a significantly lower productivity of some cultivars, in particular Tonda Gentile, compared to other cultivars was shown. Our study complements the results of Pacchiarelli et al. [43]: their data on the physiological response and productivity of the cultivars assessed at “Le Cese” is now completed with information on the susceptibility to big bud mite infestations.

The results we gathered in the first experiment can drive the establishment of future hazelnut plantations in the Viterbo province, which represents 25% of the overall Italian surface dedicated to hazelnuts [44], and in other hazelnut-growing areas similar in terms of farming conditions. The aim is in fact to achieve the highest possible quantity and quality of marketable products, while minimising the impact of agrochemicals on agroecosystems. In the area under observation, Tonda Gentile Romana is the prevalent cultivar, while Nocchione serves as the pollinator [45]. Our results showed that Tonda Gentile Romana is moderately susceptible to the action of *P. avellanae*, while Nocchione is significantly more tolerant to the mite. The spread of Nocchione in new plantations could allow a reduction in the impact of this mite on hazelnut production in the area. A proper selection of the cultivar at the establishment of the orchard is a fundamental aspect, as it can have significant effects on the sustainability of the crop production. This is directly connected with the second pillar of this study, namely the effect of orchard management on *P. avellanae* outbreaks.

The most interesting aspect that emerged from the second experiment we carried out is the significantly higher number of big buds induced by *P. avellanae* (both in the 2023 and 2024 growing season) in IPM and organic orchard management with respect to the renaturalised environment. The intensification of the production, which is a practise in IPM and organic management, leads to a reduction in biodiversity, mostly because of the use of agrochemicals and numerous destructive agronomical practises [46,47]. As a consequence, there might be an increase in insect pest and disease outbreaks, endorsed by the reduction in the density of natural enemies present in the environment [48,49]. Furthermore, the surrounding environment has a direct influence on the microbiome biodiversity within the plant tissue, which is known to be strictly involved in both tolerance/susceptibility mechanisms [50,51,52]. This aspect was already explored in a parallel study carried out at the “Le Cese” experimental field [15], where buds’ and big buds’ microbiome compositions have been investigated. The cited study assessed a higher biodiversity level in the big buds than in the healthy ones and also a different diversity level among hazelnut cultivars [15]. *Phytoptus avellanae* spends a significant part of the year inside the colonised buds [20]. Its population density is partially controlled by several species of predatory mites belonging to the Phytoseiidae family (Acari: Mesostigmata), which are naturally present in hazelnut orchards [53]. However, the use of agrochemicals to control *P. avellanae*, as well as other harmful insect pests such as the hazelnut weevil, stink bugs, or bacterial and fungal diseases, has a significant impact on predatory mites [54]. This fact is even more emphasised if treatments are carried out during the phase of migration, when *P. avellanae* specimens leave big buds to infest healthy buds at the beginning of the growing season. The composition and abundance of natural enemies can drastically change as a response to pesticide use, thus affecting their control efficiency on *P. avellanae* populations.

Organic farming is subject to very strict rules on the use of chemicals: both synthetic pesticides and fertilisers are banned to maintain high levels of biodiversity. What emerged from our results is that even organic management can lead to increased infestations of *P. avellanae*. This may be due to the effects of the few authorised active ingredients, such as neem oil and pyrethrin, or the side effects of sulphur-containing fertilisers or fungicides on the big bud mite’s natural enemies [55,56]. This fact is supported by the situation assessed in the renaturalised orchards, where the biodiversity level is assumed to be higher than in the IPM and organic environments and there is a total absence of human intervention as well. The ecosystem in this case is more stable, and there is an equilibrium such that the pool of natural enemies can maintain the extremely low level of infestation by *P. avellanae*.

The use of synthetic pesticides is, in some circumstances, indispensable; for example, they are used to contrast invasive alien species populations. However, the shift towards more sustainable management of agricultural ecosystems, increasingly seen as a social and political necessity, could activate a virtuous cycle leading to a lower impact of harmful agents on crops, as well as a decrease in environmental and health risks [57]. Our work is included in this framework, as it provides relevant information on how current agronomic practises affect the infestation trends of one of the main pests for hazelnut cultivations.

## Figures and Tables

**Figure 1 insects-15-00740-f001:**
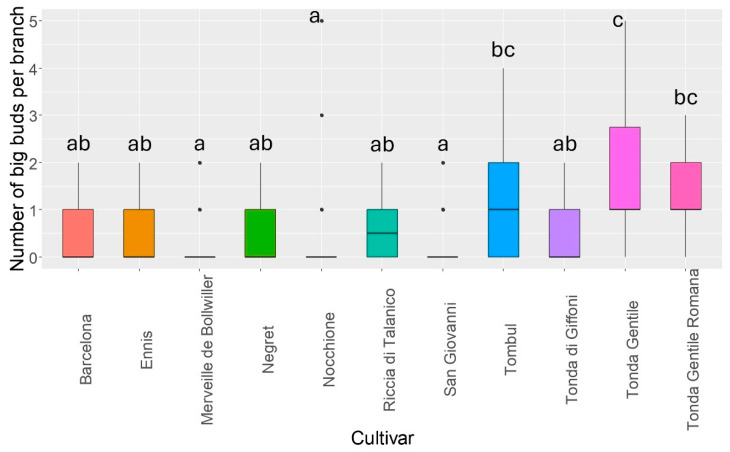
Infestation level of the 11 cultivars at “Le Cese” varietal collection, over the two years of observations, 2023 and 2024. Lines inside the boxes indicate the median values and the points outside represent the outliers. The whiskers include 95% of the data. Different letters indicate significant difference between cultivars after Bonferroni post hoc test (*p* < 0.05).

**Figure 2 insects-15-00740-f002:**
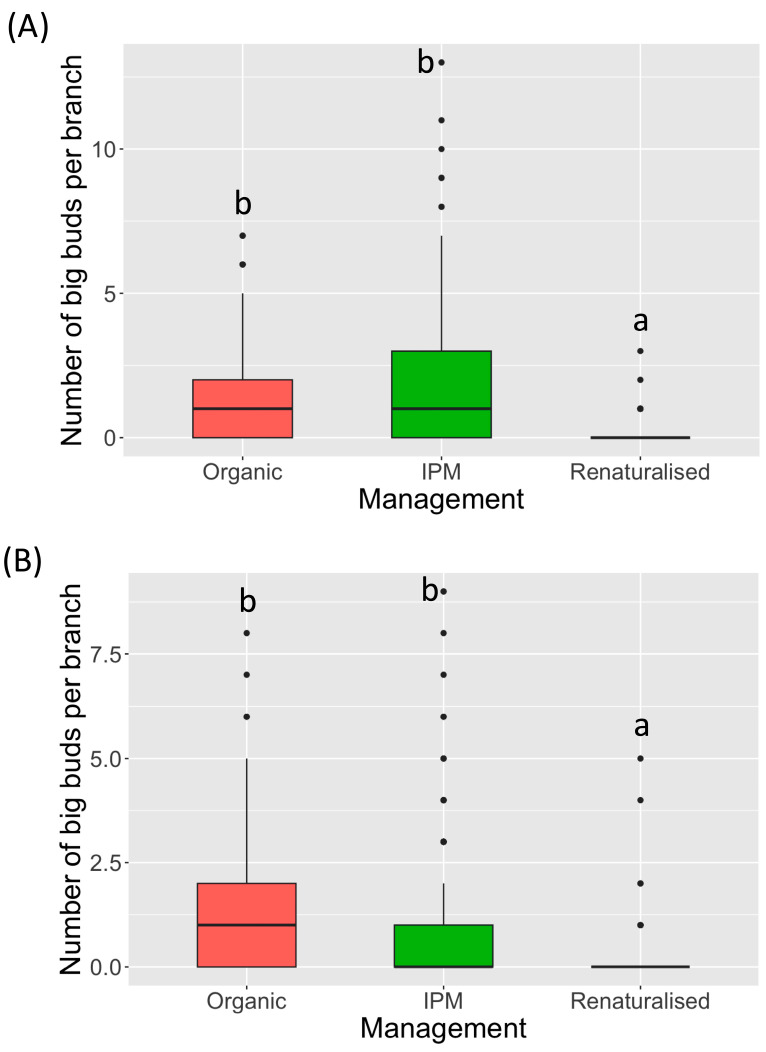
Infestation level in the orchards with organic, IPM, and renaturalised management. (**A**) growing season of 2023, (**B**) growing season of 2024. The lines inside the boxes indicate the median values and the points outside represent the outliers. The whiskers include 95% of the data. Different letters indicate significant difference between cultivars after Bonferroni post hoc test (*p* < 0.05).

**Table 1 insects-15-00740-t001:** Geographical coordinates, pest management, and year of observations of selected sites.

Site Name	Geographical Coordinates	Orchard Management	Growing Season of Observation
Calcata 1	42°12′34.2″ N 12°25′07.2″ E	Renaturalised	2023 and 2024
Calcata 2	42°13′13.3″ N 12°25′12.9″ E	Renaturalised	2023 and 2024
Calcata 3	42°13′26.8″ N 12°25′11.5″ E	Renaturalised	2023 and 2024
Ischia di Castro	42°32′47.5″ N 11°47′36.1″ E	Organic	2023 and 2024
Vetralla 1	42°17′52.7″ N 12°05′58.4″ E	Organic	2024
Bassano Romano	42°13′59.8″ N 12°10′46.4″ E	Organic	2023 and 2024
Sutri 1	42°13′26.8″ N 12°15′43.9″ E	Organic	2023
Ronciglione 1	42°16′44″ N 12°14′51″ E	Organic	2024
Ronciglione 2	42°15′34.7″ N 12°15′50.3″ E	Organic	2024
Vetralla 2	42°18′37″ N 12°05′35″ E	Organic	2024
Capranica 1	42°17′17.7″ N 12°07′48.1″ E	IPM	2023 and 2024
Capranica 2	42°17′41.6″ N 12°07′25.3″ E	IPM	2023 and 2024
Sutri 2	42°15′56.7″ N 12°13′44.5″ E	IPM	2023 and 2024
Vitorchiano	42°27′41″ N 12°11′14″ E	IPM	2024
Capranica 3	42°16′06″ N 12°07′33″ E	IPM	2024
Capranica 4	42°15′39.9″ N 12°08′26.9″ E	IPM	2024

**Table 2 insects-15-00740-t002:** Country and region of origin of hazelnut cultivars tested at “Le Cese” germplasm collection.

Cultivar	Country (and Region) of Origin ^1^
Merveille de Bollwiller	France
Tonda Gentile Romana	Italy (Latium)
Tombul	Turkey
Barcelona	Spain
Negret	Spain
Tonda Gentile	Italy (Piedmont)
Ennis	USA (Oregon)
Tonda di Giffoni	Italy (Campania)
San Giovanni	Italy (Campania)
Nocchione	Italy (Latium)
Riccia di Talanico	Italy (Campania)

^1^ The region is indicated only if the information is available.

## Data Availability

All the scripts and the raw datasets to fully reproduce the results of this study are publicly available at https://github.com/lucaros1190/Phytoptus2024LeCese (accessed on 1 September 2024).

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
