# Peer review of "Can Pest Management and Cultivar Affect Phytoptus avellanae Infestations on Hazelnut?"

_insects, 2024, doi:10.3390/insects15100740_

Round 1
Reviewer 1 Report
Comments and Suggestions for Authors
Contarini et al. report the results of a two-year study on the presence of mite infestations in hazelnut cultivars and in different pest management systems in Italy.
Main points.
My only main concern was the definition of a replicate in their study and possible spatial correlation in the distribution of replicates in their cultivar study. This was unclear to me and needs to be clearly explained.
The Discussion is rather long and focused on general issues in hazelnut production (I suggest a more tightly focussed text).
I have written suggestions and numbered points on a scanned copy of the manuscript.
Numbered points (see scanned file).
Title – did you really address the issue of "cultivar diversity" on mite infestation? I think not. Cultivar and pest management effects on mite infestation, yes.
1. The simple summary only describes the problem, not any of the results or conclusions of the study.
2. This statement is untrue. There are a diversity of phytosanitary products registered for use in organic production (Bt, fungi, spinosad, neem, etc.).
3. L.63 – change institutions worldwide to "phytosanitary authorities".
4. Please mention the presence of ground cover at each of the study sites.
5. I think it is clearer to use the word "organic" throughout the manuscript (it's not hard to write a seven letter word). The abbreviation "BIO" is unnecessary.
6. The same applies to the word "renaturalised" – please delete the abbreviation REN throughout the manuscript for improved readability.
7. Please explain replication in your study. This was unclear to me. Were replicates spatially correlated (e.g three trees in a row?).
8. Please explain your selection criteria - how or on what basis were trees selected for the study?
9. Please delete this text and just cite Table A1.
10. Please explain the meaning of the box plot. What do horizontal lines, boxes, whiskers and dots indicate in Fig 1, Fig 3.
11. Figure 2 shows a difference between years, which is to be expected in practically all studies on invertebrates. This figure conveys little information of importance and I suggest that it be deleted.
12. You are making statements based on the fluctuations observed in IPM and organic systems based on just 2 years of data. I suggest that if you had a much longer data set (10 – 20 years) your assertions may be rather different. You should perhaps be more cautious in your assertions based on such a short sampling period.
13. Delete text on general global trends (not relevant to your study).
14. The Discussion is somewhat long and general, rather than focusing on the specific issues raised by your study. I think it could be reduced in length and more tightly focused.
15. Many f the references are missing information such as volume or page numbers of the cited articles (I only revised the first page).

Comments on the Quality of English LanguageMinor editing.
Author Response
Response to Reviewer 1
Reviewer 1:
Contarini et al. report the results of a two-year study on the presence of mite infestations in hazelnut cultivars and in different pest management systems in Italy.
Response:
Dear Reviewer 1,
Thank you very much for your time dedicated to the revision of our manuscript, as well as for the helpful and interesting comments provided with the revision. We have carefully addressed all the corrections suggested, with the hope to have sufficiently increased the quality of the manuscript.
We renew our availability for any further questions or corrections, if needed.
Thank you again,
Mario Contarini and Stefano Speranza, on behalf of the authors.
Main points.
Reviewer 1:
My only main concern was the definition of a replicate in their study and possible spatial correlation in the distribution of replicates in their cultivar study. This was unclear to me and needs to be clearly explained.
Response:
Thank you for this comment. Unfortunately, we were forced to consider consecutive plants in the germplasm collection orchard, as we could not choose differently. We are aware that this is a potential limitation in the experimental scheme, but it depends on how the orchard was planted. We have better clarified this aspect into the revised text, so that the reader is well aware about this point.
Reviewer 1:
The Discussion is rather long and focused on general issues in hazelnut production (I suggest a more tightly focussed text).
Response:
Thank you for this comment. During the revision we followed the suggestion and we shortened the discussion, being more focused on the results.
Reviewer 1:
I have written suggestions and numbered points on a scanned copy of the manuscript.
Response:
Thank you for the suggestions. The comments/corrections indicated in the PDF have been included in the revision.
Reviewer 1:
Numbered points (see scanned file).
Response:
Thank you for these suggestions. Everything has been included in the revised version of the manuscript.
Reviewer 1:
Title – did you really address the issue of "cultivar diversity" on mite infestation? I think not. Cultivar and pest management effects on mite infestation, yes.
Response:
Thank you for this comment. We have removed the word “diversity” from the title, so that the research question is more coherent with the results obtained.
Reviewer 1:
- The simple summary only describes the problem, not any of the results or conclusions of the study.
Response:
Thank you for this comment. The simple summary has been prepared according to the guidelines of the Journal: it should provide an overview of the problem to indicate the context of the research, while being general. The details indicated in your comment are part of the abstract, where of course there should be an overview of the study (including results and conclusions).
Reviewer 1:
- This statement is untrue. There are a diversity of phytosanitary products registered for use in organic production (Bt, fungi, spinosad, neem, etc.).
Response:
Thank you for this comment. We have revised this sentence by referring to “synthetic pesticides” instead of “phytosanitary treatments”.
Reviewer 1:
- L.63 – change institutions worldwide to "phytosanitary authorities".
Response:
Thank you for this comment. We have corrected this part of the text accordingly.
Reviewer 1:
- Please mention the presence of ground cover at each of the study sites.
Response:
Thank you for this comment. In the revised text we added information about the ground cover of the experimental orchards.
Reviewer 1:
- I think it is clearer to use the word "organic" throughout the manuscript (it's not hard to write a seven letter word). The abbreviation "BIO" is unnecessary.
Response:
Thank you for this comment. In the revised manuscript we removed the abbreviation “BIO”.
Reviewer 1:
- The same applies to the word "renaturalised" – please delete the abbreviation REN throughout the manuscript for improved readability.
Response:
Thank you for this comment. We have corrected the text accordingly.
Reviewer 1:
- Please explain replication in your study. This was unclear to me. Were replicates spatially correlated (e.g three trees in a row?).
Response:
Thank you for this comment. In the germplasm collection we were forced to inspect consecutive plants (three trees in a row) as the different varieties were placed in this way.
In the second experiment, instead, plants were selected randomly avoiding, as much as possible, spatial correlations. As already discussed before, the experimental scheme was set to reduce as much as possible correlations or undesired effects due to favourable/unfavourable development conditions for the mite. For the sake of clarity, we have better explained these details, pointing out the strengths and the weak points of the experimental schemes. We hope that now the text is sufficiently clearer.
Reviewer 1:
- Please explain your selection criteria - how or on what basis were trees selected for the study?
Response:
Thank you for this comment. As already stated and discussed, plants were randomly selected into each orchard. The criteria we adopted were the following:
- Plants should be far enough to avoid spatial correlations
- Plants should be far enough from the field borders
The fields were big enough to warrant these conditions. Moreover, repeating the monitoring on different farms (obviously not neighbouring) contributed to decrease the spatial correlation.
The situation was different in the germplasm collection orchard, as plants belonging to different varieties were placed consecutively in a line.
Reviewer 1:
- Please delete this text and just cite Table A1.
Response:
Thank you for this suggestion. We have corrected the text accordingly.
Reviewer 1:
- Please explain the meaning of the box plot. What do horizontal lines, boxes, whiskers and dots indicate in Fig 1, Fig 3.
Response:
Thank you for this comment. We have corrected the figure captions accordingly.
Reviewer 1:
- Figure 2 shows a difference between years, which is to be expected in practically all studies on invertebrates. This figure conveys little information of importance and I suggest that it be deleted.
Response:
Thank you for this comment. Although it is not true that differences in infestation levels between two consecutive years “is to be expected”, we removed figure 2, as we agree that it is not adding more information than what is already written in the main text.
Reviewer 1:
- You are making statements based on the fluctuations observed in IPM and organic systems based on just 2 years of data. I suggest that if you had a much longer data set (10 – 20 years) your assertions may be rather different. You should perhaps be more cautious in your assertions based on such a short sampling period.
Response:
Thank you for this comment. We have rephrased this part of the text by just referring to what we observed in the two years.
Reviewer 1:
- Delete text on general global trends (not relevant to your study).
Response:
Thank you for this comment. We have corrected the manuscript accordingly.
Reviewer 1:
- The Discussion is somewhat long and general, rather than focusing on the specific issues raised by your study. I think it could be reduced in length and more tightly focused.
Response:
Thank you for this comment. During the revision we have shortened and re-focused the discussion.
Reviewer 1:
- Many of the references are missing information such as volume or page numbers of the cited articles (I only revised the first page).
Response:
Thank you for this comment. We have revised the reference list accordingly.
Reviewer 2 Report
Comments and Suggestions for Authors
This is a very interesting study. The text is nicely written, however, I suggest a detailed revision of the English language to enhance the formality of the text and overall presentation of the findings. It is also suggested to adjust the methodology for clarification purposes and reconstruct the discussion section to enhance the flow and overall readability of the text.
Line 30: Please avoid the use of acronyms unless their meaning is explained.
Line 40: Please include the full authorities of a species (order: family) throughout the text at the first time mentioned.
Lines 40-52: I suggest combining this part into a single paragraph, as well as lines 53-64.
Line 66: Did you mean “sensu lato”?
Line 67: Consider “a substantial obstacle”.
Line 71: Consider “delayed plant development”.
Line 74: Consider “the main cause of production losses”.
Line 75-80: Please add reference(s) here.
Lines 95-106: This belongs to the methodology section. Please avoid elaborating on the methodology. The authors could expand lines 90-94 as a concluding paragraph of the introduction, shortly providing the gaps in the literature and the scope of the study. There is no need to describe the experiments here.
Lines 123-130: Which cultivars were present in each environment (IPM, BIO, and REN).
Line 140: Please add a reference regarding the identification keys.
Lines 212-218: I suggest using “higher/lower infestation level” instead of “different” here.
Figures 2 and 3: Please, remove the titles from the figures. The legends fully explain the figures.
Lines 258-312: This part of the discussion could be improved by reorganizing it as follows: a) Contrast the current findings with previous research to contextualize the results, b) Provide a more detailed explanation of the factors behind the observed differences in susceptibility (currently in lines 291-295), c) Summarize the current status in Italy and Turkey to provide relevant context, d) Offer specific recommendations based on the comparisons and context provided (currently lines 307-312).
Lines 265-266: Please add a reference.
Line 271: Please avoid repetitions on the organoleptic characteristics. To this end, you could merge this paragraph with the previous one.
Lines 295-298: I do not understand. Do you mean that the current research adds information on the damage caused by big bug mites? Please rephrase.
Lines 338-341: In the current study, IPM and BIO did not yield significant differences. Please mention some factors that could influence BIO management as in IPM in the previous paragraph. Also, were the cultivars different in each environment? If yes, how did this affect the results?
Lines 346-363: I recommend replacing this paragraph with lines 246-257 as a take-home message of the findings of this study and their importance. The authors may introduce lines 346-363 at the beginning of the introduction with the appropriate modifications.
Best of luck!
Comments on the Quality of English LanguageAlthough the text is decently written, I recommend to the authors that they conduct a comprehensive revision of the English language to enhance the formality and overall presentation.
Author Response
Response to Reviewer 2
Reviewer 2:
This is a very interesting study. The text is nicely written, however, I suggest a detailed revision of the English language to enhance the formality of the text and overall presentation of the findings. It is also suggested to adjust the methodology for clarification purposes and reconstruct the discussion section to enhance the flow and overall readability of the text.
Response:
Dear Reviewer 2,
Thank you very much for your time dedicated to revise our manuscript, as well as for the helpful comments and suggestions provided with the revision. We have carefully addressed all the corrections suggested, with the hope to have sufficiently increased the quality of the manuscript.
We renew our availability for any further questions or corrections, if needed.
Thank you again,
Mario Contarini and Stefano Speranza, on behalf of the authors.
Reviewer 2:
Line 30: Please avoid the use of acronyms unless their meaning is explained.
Response:
Thank you for this suggestion. During the revision we paid attention to remove the unnecessary acronyms.
Reviewer 2:
Line 40: Please include the full authorities of a species (order: family) throughout the text at the first time mentioned.
Response:
Thank you for this suggestion. During the revision we added the full authorities throughout the text on the first time species were mentioned.
Reviewer 2:
Lines 40-52: I suggest combining this part into a single paragraph, as well as lines 53-64.
Response:
Thank you for this suggestion. During the revision we addressed this correction.
Reviewer 2:
Line 66: Did you mean “sensu lato”?
Response:
Thank you for this comment. According to Latin, the correct form is “lato sensu”, even if the form “sensu lato” is accepted as well in English.
Reviewer 2:
Line 67: Consider “a substantial obstacle”.
Response:
Thank you for this suggestion. We have changed this part of the text accordingly.
Reviewer 2:
Line 71: Consider “delayed plant development”.
Response:
Thank you for this suggestion. We have changed this part of the text accordingly.
Reviewer 2:
Line 74: Consider “the main cause of production losses”.
Response:
Thank you for this suggestion. We have changed this part of the text accordingly.
Reviewer 2:
Line 75-80: Please add reference(s) here.
Response:
Thank you for this comment. During the revision we added references to support the statement.
Reviewer 2:
Lines 95-106: This belongs to the methodology section. Please avoid elaborating on the methodology. The authors could expand lines 90-94 as a concluding paragraph of the introduction, shortly providing the gaps in the literature and the scope of the study. There is no need to describe the experiments here.
Response:
Thank you for this comment. The scope of this paragraph was to describe the goals of our study while summarising what we effectively did. We understand that this is a personal taste, as in papers introduction can just end with the goals (as you kindly suggested) or with a brief summary of what the reader is going to read (as we did). Despite we deeply considered this change, during the revision we decided to address the suggestions of the other reviewers on this paragraph, without any shortening. We hope that the revised form of the paragraph better suits your expectations.
Reviewer 2:
Lines 123-130: Which cultivars were present in each environment (IPM, BIO, and REN).
Response:
Thank you for this comment. We have expanded this part of the text accordingly, by providing the requested information.
Reviewer 2:
Line 140: Please add a reference regarding the identification keys.
Response:
Thank you very much for this comment. We have integrated the text with the requested citation.
Reviewer 2:
Lines 212-218: I suggest using “higher/lower infestation level” instead of “different” here.
Response:
Thank you for this suggestion. We corrected this part of the text accordingly.
Reviewer 2:
Figures 2 and 3: Please, remove the titles from the figures. The legends fully explain the figures.
Response:
Thank you for this comment. We have modified the figures accordingly.
Reviewer 2:
Lines 258-312: This part of the discussion could be improved by reorganizing it as follows: a) Contrast the current findings with previous research to contextualize the results, b) Provide a more detailed explanation of the factors behind the observed differences in susceptibility (currently in lines 291-295), c) Summarize the current status in Italy and Turkey to provide relevant context, d) Offer specific recommendations based on the comparisons and context provided (currently lines 307-312).
Response:
Thank you for these suggestions. We have reorganised this part of the discussion accordingly, including the suggestions of the other two reviewers as well.
Reviewer 2:
Lines 265-266: Please add a reference.
Response:
Thank you for this comment. During the revision we have provided the requested citation.
Reviewer 2:
Line 271: Please avoid repetitions on the organoleptic characteristics. To this end, you could merge this paragraph with the previous one.
Response:
Thank you for this suggestion. The two paragraphs were merged during the revision.
Reviewer 2:
Lines 295-298: I do not understand. Do you mean that the current research adds information on the damage caused by big bug mites? Please rephrase.
Response:
Thank you for this comment. Yes, the idea behind this sentence was to remark that Pacchiarelli et al. provided important information on the physiology and the productivity of the cultivars at “Le Cese”. Our study, instead, complements the information with the susceptibility of the cultivars to mites. During the revision we have better clarified this aspect by rephrasing the sentence.
Reviewer 2:
Lines 338-341: In the current study, IPM and BIO did not yield significant differences. Please mention some factors that could influence BIO management as in IPM in the previous paragraph. Also, were the cultivars different in each environment? If yes, how did this affect the results?
Response:
Thank you for these questions. We have revised this part of the manuscript accordingly, providing the requested information.
Reviewer 2:
Lines 346-363: I recommend replacing this paragraph with lines 246-257 as a take-home message of the findings of this study and their importance. The authors may introduce lines 346-363 at the beginning of the introduction with the appropriate modifications.
Response:
Thank you for this comment. We have reshaped this part of the discussion taking into account those suggestions as well.
Reviewer 2:
Best of luck!
Response:
Thank you again for your time and for the helpful comments on our work! We hope that the revised version of the manuscript better suits your expectations.
Reviewer 2:
Comments on the Quality of English Language
Although the text is decently written, I recommend to the authors that they conduct a comprehensive revision of the English language to enhance the formality and overall presentation.
Response:
Thank you for this suggestion. During the revision we have paid particular attention to this aspect as well.
Reviewer 3 Report
Comments and Suggestions for Authors
Dear Authors,
I have carefully read your manuscript. The topic is original and pertains to a crop that is rapidly expanding in Italy and beyond.
However, I have noticed several inaccuracies in the experimental design that require attention. These issues could necessitate a thorough revision of the data collected and the results obtained from their analysis.
Below are my main comments and suggestions, which should be taken into consideration for a reassessment of the work.
-
Line 37: I recommend expanding the keywords to improve the manuscript's visibility in research databases.
-
Line 111: Clearly specify whether "Le Cese" is an experimental field, a germplasm conservation center, a farm, etc., without referring to the description in a bibliographic reference. This is a study containing direct experimental data, and the research areas must be described by the authors themselves, with potential integration from other works. Including a map or image identifying the study area would also be valuable.
-
Lines 131-136: Specify if the management of the hazelnut orchards follows the integrated cultivation guidelines provided by the Lazio region (with a link or reference to the document if possible).
-
Lines 145-146, Table 1: I have some concerns about the analysis of incomplete data for both years across the different study sites. The authors should either provide a robust justification for using all available data despite the missing annual data for some sites or supplement the information.
-
Line 148: Is the monitoring limited to just the month of January, the coldest period of the year and during the plants' vegetative dormancy? I believe the authors should consider these points and provide adequate justifications.
-
Line 155: In a study of this kind, the full monitoring protocol needs to be reported, not just cited.
-
Lines 156-159: Have the authors considered whether the different plant exposures could affect the biological cycle of the pests and their distribution on the plants?
-
Line 238, Figure: Although indicated with different letters, the reader does not detect statistically significant differences between the two years in the graph.
-
Line 242, Figure 3: Were the data from sites observed only in 2024 included in the following analysis? If so, the comparison between the two years is scientifically irrelevant.
-
Line 252: Replace Phytoptus avellanae with P. avellanae.
-
Line 260: Replace Phytoptus avellanae with P. avellanae.
-
Line 318: To assert that agricultural management practices and pesticide use in the area have reduced biodiversity, the specific field management operations should be detailed in the experimental design.
I believe the entire discussion and conclusion section should be revised after clarifying the experimental design and, consequently, reassessing the results.
Lastly, I recommend paying close attention to the linguistic style, which currently does not meet the standards of a scientific article.
Best regards, and good luck with your work.
Comments on the Quality of English LanguageModerate editing of English language required.
Author Response
Response to Reviewer 3
Reviewer 3:
Dear Authors,
I have carefully read your manuscript. The topic is original and pertains to a crop that is rapidly expanding in Italy and beyond.
However, I have noticed several inaccuracies in the experimental design that require attention. These issues could necessitate a thorough revision of the data collected and the results obtained from their analysis.
Below are my main comments and suggestions, which should be taken into consideration for a reassessment of the work.
Response:
Dear Reviewer 3,
Thank you very much for your time dedicated to the revision of our manuscript, as well as for the helpful and interesting comments provided with the revision. We have carefully addressed all the corrections suggested, with the hope to have sufficiently increased the quality of the manuscript.
We renew our availability for any further questions or corrections, if needed.
Thank you again,
Mario Contarini and Stefano Speranza, on behalf of the authors.
Reviewer 3:
Line 37: I recommend expanding the keywords to improve the manuscript's visibility in research databases.
Response:
Thank you for this suggestion. During the revision we added some key words such as “big bud mite” and “phytosanitary management” to improve the manuscript’s visibility.
Reviewer 3:
Line 111: Clearly specify whether "Le Cese" is an experimental field, a germplasm conservation center, a farm, etc., without referring to the description in a bibliographic reference. This is a study containing direct experimental data, and the research areas must be described by the authors themselves, with potential integration from other works. Including a map or image identifying the study area would also be valuable.
Response:
Thank you for this suggestion. We modified this part of the manuscript by better explaining the characteristics of this experimental field.
Reviewer 3:
Lines 131-136: Specify if the management of the hazelnut orchards follows the integrated cultivation guidelines provided by the Lazio region (with a link or reference to the document if possible).
Response:
Thank you for this suggestion. During the revision we provided a better description of the different pest management practices under which the research was carried out, following the suggestions proposed.
Reviewer 3:
Lines 145-146, Table 1: I have some concerns about the analysis of incomplete data for both years across the different study sites. The authors should either provide a robust justification for using all available data despite the missing annual data for some sites or supplement the information.
Response:
Thank you for this comment. Our idea, while planning the experimentation, was to repeat the measurements in the same hazelnut orchards for both years. However, this has not been possible, as factors that did not depend on our decisions led us to do some on-the-go modifications. For instance, one of the organic orchards switched to IPM, leading us to find a replacement. For the second year, instead, we decided to expand the number of fields monitored with the conviction that a higher number of fields would have provided an even more accurate overview of the impact that management practices have on P. avellanae infestations. In all honesty and clarity, this change did not include any correlation in the dataset and/or unbalance, as the statistical methods to analyse the data were chosen properly to avoid any bias. We can consider the experimental trial as a multi-layer random sampling: we have randomly selected the branches in each plant, plants into the field and the orchards in the whole area. The experimental trials were devoted to reducing correlations and undesired effects as much as possible. We hope to have provided a sufficiently “robust” justification; we moreover extended this part of the text to be as clear as possible with the readers.
Reviewer 3:
Line 148: Is the monitoring limited to just the month of January, the coldest period of the year and during the plants' vegetative dormancy? I believe the authors should consider these points and provide adequate justifications.
Response:
Thank you for this comment. During the revision we have better explained the rationale behind the choice to carry out the monitoring in January. We wish to point out, moreover, that this is a standard practice for this pest on hazelnut plantations and that it has been already applied in other studies from different authors.
Reviewer 3:
Line 155: In a study of this kind, the full monitoring protocol needs to be reported, not just cited.
Response:
Thank you for this suggestion. As we explained in the previous comment, the monitoring protocol was not invented de novo, but it was kept from other studies. For this reason, we have cited the corresponding literature to give credit to the authors. The methodology, however, was described shortly afterwards and during the revision we slightly extended the set of information already present.
Reviewer 3:
Lines 156-159: Have the authors considered whether the different plant exposures could affect the biological cycle of the pests and their distribution on the plants?
Response:
Thank you for this comment. This is a very interesting point. In our previous studies (Contarini et al. 2022) we observed that the infestation is more concentrated in the central part of the canopy, avoiding the top (more or directly exposed to the sun) and the bottom (where the availability of branches and consequently of buds to infest is lower). In a certain sense, we already explored the exposure to sun, even if there are other aspects that further research should consider. For instance, it would be interesting to explore if there is a correlation between the cardinal direction and the infestation level, but this is an experiment per se. Based on the data we have, we cannot provide any information, also because it was not among the aims of this study. Thank you very much for the hint!
Reviewer 3:
Line 238, Figure: Although indicated with different letters, the reader does not detect statistically significant differences between the two years in the graph.
Response:
Thank you for this comment. As this figure is generating confusion and it is not pivotal for the study, we decided to remove it from the text. We hope that this choice has improved the clarity of the text.
Reviewer3:
Line 242, Figure 3: Were the data from sites observed only in 2024 included in the following analysis? If so, the comparison between the two years is scientifically irrelevant.
Response:
Thank you for this suggestion. Yes, the year is a factor in the analysis and for this reason the comparison between the two years makes sense and has been carried out. Based on this comment, we have revised the methodology to double check if this fact was correctly explained.
Reviewer 3:
Line 252: Replace Phytoptus avellanae with P. avellanae.
Response:
Thank you for this suggestion. This is a purely stylistic comment that to date cannot be referred to any standards. As far as we know, at the beginning of the sentence the genus name should not be abbreviated, and for this reason we did not address this correction. We think that the best option is to leave the decision to the editorial office, if the manuscript will be accepted.
Reviewer 3:
Line 260: Replace Phytoptus avellanae with P. avellanae.
Response:
Thank you for this suggestion. We have changed the text accordingly.
Reviewer 3:
Line 318: To assert that agricultural management practices and pesticide use in the area have reduced biodiversity, the specific field management operations should be detailed in the experimental design.
Response:
Thank you for this comment. Based on the comments above, we have extended the part of the methodology by complementing with the missing information. We hope that this change better supports the statements of the discussion.
Reviewer 3:
I believe the entire discussion and conclusion section should be revised after clarifying the experimental design and, consequently, reassessing the results.
Response:
Thank you for this comment. During the revision we paid particular attention to discussion and conclusion, improving the weak parts and removing the redundancies. We hope that the revised discussion is clearer.
Reviewer 3:
Lastly, I recommend paying close attention to the linguistic style, which currently does not meet the standards of a scientific article.
Response:
Thank you for this comment. During the revision we paid particular attention to the English style, also thanks to the suggestions received.
Reviewer 3:
Best regards, and good luck with your work.
Response:
Thank you again for your time and for the helpful comments and suggestions. We hope that the revised version of the manuscript better suits your expectations.
Round 2
Reviewer 2 Report
Comments and Suggestions for Authors
Congratulations to the authors; this article has seen great improvement, particularly in terms of clarifying the methodology. There are only a few points to address, listed below.
Line 30: Please use IPM in full at the first time mentioned, i.e., “Integrated Pest Management (IPM)”
Lines 53-65: I suggest combining this part into a single paragraph.
Line 146: Please use “cultivar” instead of “cv” for consistency throughout the text.
Figure 2: Please, remove the titles from the figure. The legends fully explain the figures A and B.
Lines 285-287: Any examples/references?
Lines 353-354: Please add a reference here.
Comments on the Quality of English LanguageI would suggest that the authors use a writing assistant tool to improve clarity, and grammar, and polish the English language.
Author Response
Congratulations to the authors; this article has seen great improvement, particularly in terms of clarifying the methodology. There are only a few points to address, listed below.
Response
Dear Reviewer,
we would like to once again express our gratitude for the time you have dedicated and for your valuable suggestions aimed at improving the quality of the manuscript.
Line 30: Please use IPM in full at the first time mentioned, i.e., “Integrated Pest Management (IPM)”
Response:
Thank you for this comment. We added the full name.
Lines 53-65: I suggest combining this part into a single paragraph.
Response:
Thank you for this comment. We have combined the section into a single paragraph
Line 146: Please use “cultivar” instead of “cv” for consistency throughout the text.
Response:
Thank you for this comment. We have replaced cv with cultivar
Figure 2: Please, remove the titles from the figure. The legends fully explain the figures A and B.
Response:
Thank you for this comment. We modified the figure accordingly
Lines 285-287: Any examples/references?
Lines 353-354: Please add a reference here.
Response:
Thank you for the suggestions. Although we experienced some difficulties in identifying the specific lines where citations needed to be added, our review of the discussions allowed us to pinpoint sections where the bibliographic references could be included. We are confident that we have also incorporated the suggested parts.
Comments on the Quality of English Language
I would suggest that the authors use a writing assistant tool to improve clarity, and grammar, and polish the English language.
Response:
Thank you, we asked a native speaker to conduct a thorough linguistic review of the article in order to improve the grammatical quality and clarity.
Reviewer 3 Report
Comments and Suggestions for Authors
Dear Authors,
I appreciate your effort. The manuscript has certainly improved, but I am not entirely satisfied with how some of my suggestions have been addressed (e.g., the removal of Figure 2, the linguistic style in certain parts of the text, and the lack of revision in the discussion section, including the enhancement of bibliographic references, as suggested). I leave it to your discretion to review my initial report and provide a more thorough response to some of my concerns. Best of luck with your work.
Comments on the Quality of English LanguageMinor editing of English language required.
Author Response
Dear Authors,
I appreciate your effort. The manuscript has certainly improved, but I am not entirely satisfied with how some of my suggestions have been addressed (e.g., the removal of Figure 2, the linguistic style in certain parts of the text, and the lack of revision in the discussion section, including the enhancement of bibliographic references, as suggested). I leave it to your discretion to review my initial report and provide a more thorough response to some of my concerns. Best of luck with your work.
Dear Reviewer 3,
We would like to once again express our sincere gratitude for the time and effort you have dedicated, as well as for your valuable suggestions. In this second round of revision, we modified the fig 2 according to your recommendation, and we have revised the discussion by including some citations to strengthen the arguments presented. Finally, we asked a native speaker to conduct a thorough linguistic review of the manuscript. We are confident that we have now successfully incorporated your suggestions to improve the quality of the manuscript.